# CodeT5+: Open Code Large Language Models for Code Understanding and Generation

**Yue Wang**[*], **Hung Le**[*], **Akhilesh Deepak Gotmare, Nghi D.Q. Bui, Junnan Li, Steven C.H. Hoi**
Salesforce AI Research
https://github.com/salesforce/CodeT5/tree/main/CodeT5+

## Abstract

Large language models (LLMs) pretrained on vast source code have achieved prominent progress in code intelligence. However, existing code LLMs have two main limitations. First, they often adopt a specific architecture (encoder-only or decoder-only) or rely on a unified encoder-decoder network for different downstream tasks, lacking the flexibility to operate in the optimal architecture for a specific task. Secondly, they often employ a limited set of pretraining objectives which might not be relevant to some tasks and hence result in substantial performance degrade. To address these limitations, we propose "CodeT5+", a family of encoder-decoder LLMs for code in which component modules can be flexibly combined to suit a wide range of code tasks. Such flexibility is enabled by our proposed mixture of pretraining objectives, which cover span denoising, contrastive learning, text-code matching, and causal LM pretraining tasks, on both unimodal and bimodal multilingual code corpora. Furthermore, we propose to initialize CodeT5+ with frozen off-the-shelf LLMs without training from scratch to efficiently scale up our models, and explore instruction-tuning to align with natural language instructions. We extensively evaluate CodeT5+ on over 20 code-related benchmarks in different settings, including zero-shot, finetuning, and instruction-tuning. We observe state-of-the-art (SoTA) performance on various code-related tasks, and our instruction-tuned CodeT5+ 16B achieves new SoTA results of $35.0\%$ pass@1 and $54.5\%$ pass@10 on the HumanEval code generation task against other open code LLMs, even surpassing the OpenAI code-cushman-001 model.

## 1 Introduction

Large language models (LLMs) (Chen et al., 2021; Wang et al., 2021b; Nijkamp et al., 2023b) have recently demonstrated remarkable success in a broad

set of downstream tasks in the code domain (Husain et al., 2019; Lu et al., 2021; Hendrycks et al., 2021). By pretraining on massive code-based data (e.g. GitHub public data), these code LLMs can learn rich contextual representations which can be transferred to various code-related downstream tasks. However, we found that many existing models are designed to perform well only in a subset of tasks. We argue that this is mainly due to two limitations in terms of architecture and pretraining tasks.

From an architectural perspective, existing code LLMs often adopt encoder-only or decoder-only models that perform well only on certain understanding or generative tasks. Specifically, encoder-only models (Feng et al., 2020; Guo et al., 2021) are often used to facilitate understanding tasks such as text-to-code retrieval (Lu et al., 2021). For generative tasks such as code generation (Chen et al., 2021; Hendrycks et al., 2021), decoder-only models (Chen et al., 2021; Nijkamp et al., 2023b) often demonstrate stronger performance. However, these decoder-only models are often not ideal for understanding tasks such as detection tasks compared to encoder-only models (Nijkamp et al., 2023a).

Besides, several models have adopted more unified encoder-decoder architectures (Wang et al., 2021b; Ahmad et al., 2021) to adapt to different types of tasks. While these models can support both understanding and generative tasks, they still suffer from suboptimal performance on certain tasks. Guo et al. (2022) found that encoder-decoder models fail to beat (state-of-the-art) SoTA encoder-only or decoder-only baselines on retrieval and code completion tasks respectively. This shortfall is due to the limitation of the single-module architecture generally adapted to all tasks. In summary, prior approaches are not designed with compositionality such that individual components can be activated to better suit different types of downstream tasks.

From a learning objective perspective, current models employ a limited set of pretraining tasks.

---

[*]Equal contribution. Corresponding authors: {wang.y, hungle, shoi}@salesforce.com

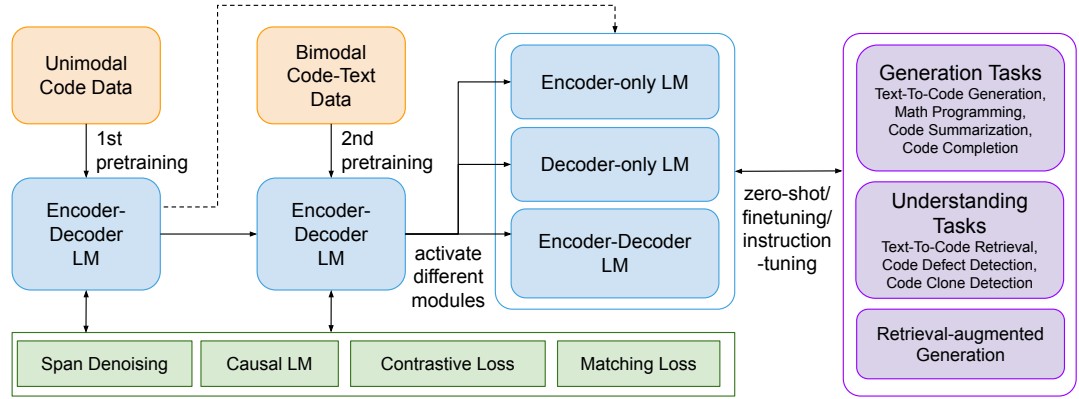

Figure 1: An overview of our CodeT5+, a family of code LLMs for code understanding and generation.

These tasks can lead to performance degrade on certain downstream tasks due to the discrepancy between the pretraining and finetuning stage. For instance, T5-based models such as (Wang et al., 2021b) are often trained with a span denoising objective. However, in downstream tasks such as code generation (Chen et al., 2021; Hendrycks et al., 2021), most SoTA models are pretrained with a next-token prediction objective which auto-regressively predicts a program token by token. Furthermore, many models are not trained to learn contrastive code representations that are vital for understanding tasks such as text-to-code retrieval. Although recent attempts (Guo et al., 2022; Wang et al., 2021a) introduce a contrastive learning task to alleviate this issue, these approaches ignore the fine-grained text-code cross-modal alignments.

To address the above limitations, we propose "CodeT5+", a new family of encoder-decoder code foundation LLMs for a wide range of code understanding and generation tasks (see Fig. 1 for an overview). Despite being an encoder-decoder based model, our CodeT5+ can flexibly operate in encoder-only, decoder-only, and encoder-decoder modes to suit different downstream applications. Such flexibility is enabled by our proposed pretraining tasks, which include span denoising and causal language modeling (CLM) tasks on code data and text-code contrastive learning, matching, and CLM tasks on text-code data. We found that such a wide set of pretraining tasks can help learn rich representations from both code and text data, and bridge the pretrain-finetune gap in various downstream applications. Besides, we show that the integration of the matching task with contrastive learning is crucial to capture the fine-grained text-code alignments and improve retrieval performance.

Furthermore, we scale up the model size of CodeT5+ with a compute-efficient pretraining strategy by leveraging off-the-shelf code LLMs to initialize the components of CodeT5+. Specifically, we employ a "shallow encoder and deep decoder" architecture (Li et al., 2022b), where both encoder and decoder are initialized from pretrained checkpoints and connected by cross-attention layers. We freeze the deep decoder LLM and only train the shallow encoder and cross-attention layers, largely reducing the number of trainable parameters for efficient tuning. Finally, recent work in the NLP domain (Taori et al., 2023; Wang et al., 2022; Ouyang et al., 2022) inspired us to explore CodeT5+ with instruction tuning to better align the models with natural language instructions.

We extensively evaluate CodeT5+ on over 20 code-related benchmarks under various settings, including zero-shot, finetuning, and instruction-tuning. Results show that CodeT5+ yields substantial performance gains on many downstream tasks compared to their SoTA baselines, e.g., 8 text-to-code retrieval tasks (+3.2 avg. MRR), 2 line-level code completion tasks (+2.1 avg. Exact Match), and 2 retrieval-augmented code generation tasks (+5.8 avg. BLEU-4). In 2 math programming tasks on MathQA and GSM8K benchmarks (Austin et al., 2021; Cobbe et al., 2021), CodeT5+ models of below billion-parameter sizes significantly outperform many LLMs of up to 137B parameters. Particularly, in the zero-shot text-to-code generation task on HumanEval benchmark (Chen et al., 2021), our instruction-tuned CodeT5+ 16B sets new SoTA results of 35.0% pass@1 and 54.5% pass@10 against other open code LLMs, even surpassing the closed-source OpenAI code-cushman-001 model. Finally, we showcase that CodeT5+ can be seamlessly adopted as a semi-parametric retrieval-augmented generation system.

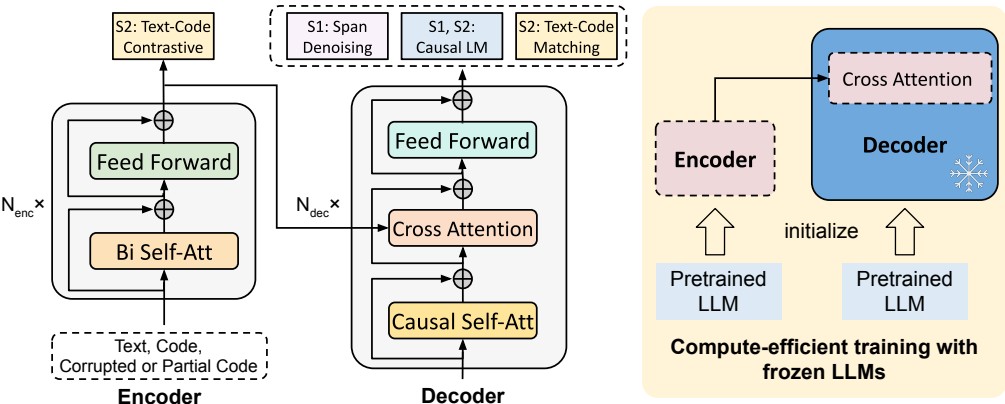

Figure 2: Model architecture of CodeT5+. S1: first stage pretraining with unimodal code data, S2: second stage pretraining with bimodal code-text data. The diagram on the right shows our proposed compute-efficient training with frozen code LLMs to scale up the model. We employ a "shallow encoder and deep decoder" architecture and only keep the small encoder and the cross-attention layers trainable while freezing the deep decoder LLM.

## 2 Related Work

Following the success of LLMs such as BERT (Devlin et al., 2019) and GPT (Radford et al., 2019) in natural language processing (NLP), recent years witness a surge of research work of LLMs in the code domain, leading to new SoTA results on various code-related tasks. Typically, code LLMs can be categorized into three architectures: encoder-only models (Feng et al., 2020), decoder-only models (Chen et al., 2021; Nijkamp et al., 2023b), and encoder-decoder models (Ahmad et al., 2021; Wang et al., 2021b). For encoder-only and decoder-only models, they are often ideal for either understanding tasks such as code retrieval (Husain et al., 2019) or generation tasks such as code synthesis (Chen et al., 2021) respectively. For encoder-decoder models, they can be adapted to both code understanding and generation but do not always achieve better performance (Wang et al., 2021b). In this work, we propose a new family of encoder-decoder code LLMs "CodeT5+" that can flexibly operate in various modes, including encoder-only, decoder-only, and encoder-decoder models.

Prior code LLMs are also limited by their pretraining tasks, which are not perfect to transfer the models to some downstream tasks. For instance, T5-based models such as (Wang et al., 2021b) pretrained with span denoising objective are not ideal for auto-regressive generation tasks like next-line code completion (Guo et al., 2022), as these models are trained to recover short spans of limited lengths rather than a whole program.[1] Inspired by recent

advances in NLP research (Tay et al., 2022; Soltan et al., 2022), we explore to combine span denoising with CLM tasks to improve the model with better causal generation capability (Le et al., 2022). Additionally, most models do not have specific pretraining tasks (e.g. contrastive learning) to facilitate the learning of contextual representations that can distinguish code samples of different semantics. This can lead to suboptimal performance on code understanding tasks like code retrieval (Husain et al., 2019). In light of this, we include a contrastive learning task to learn better unimodal representations and a matching task to learn richer bimodal representations, which has been shown helpful in vision-language retrieval tasks (Li et al., 2021).

## 3 CodeT5+: Open Code LLMs

We develop CodeT5+, a new family of open code LLMs for code understanding and generation tasks (see Fig. 1 for an overview and more architecture/pretraining details in Fig. 2 and Fig. 3). Based on the encoder-decoder architecture (Wang et al., 2021b), CodeT5+ is enhanced with the flexibility to operate in various modes for different downstream tasks through our proposed mixture of pretraining objectives, which are performed on two stages of pretraining on unimodal (Sec. 3.1) and bimodal data (Sec. 3.2). We found that this stage-wise training approach can efficiently expose our models to more diverse data to learn rich contextual representations. Finally, we explore initializing CodeT5+ with off-the-shelf code LLMs to efficiently scale up the model without training from scratch (Sec. 3.3).

---

[1]Tabachnyk and Nikolov (2022); Fried et al. (2022) demonstrated using encoder-decoder models for infilling-style code completion, in which code context after the cursor is provided.

Such code completion setting is not the focus of this work.

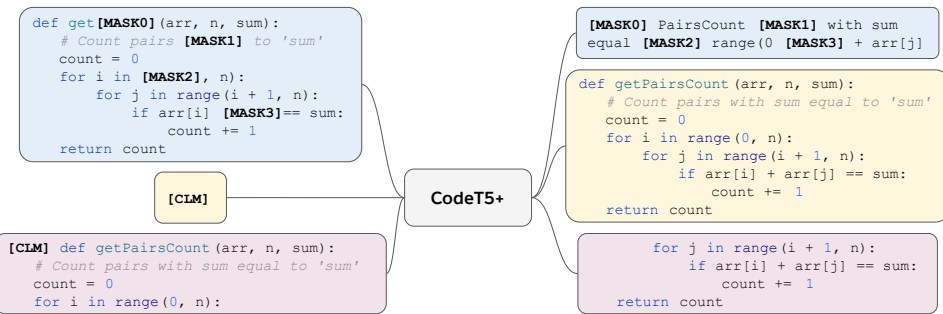

Figure 3: **Self-supervised pretraining on code data:** we pretrain CodeT5+ on code data using a mixture of tasks: (i) span denoising (Top); (ii) decoder-only causal LM (Middle); and (iii) Seq2Seq causal LM (Bottom). This mixture of tasks lets the models learn meaningful representations of code contexts and recover missing information at different levels: code spans, partial programs, and complete programs.

## 3.1 Unimodal Pretraining on Code Data

In the first stage, we pretrain CodeT5+ on large-scale code unimodal data, which can be obtained from open-source platforms like GitHub. Although such data also contain texts such as user-written code comments, we denote unimodal data to distinguish them with bimodal data of text-code pairs in the second pretraining stage. In this stage, we pretrain the model from scratch using a mixture of span denoising and CLM tasks as shown in Fig. 3. These tasks enable the model to learn to recover code contexts at different scales: code spans, partial programs, and complete programs.

**Span Denoising.** Similar to T5 (Raffel et al., 2020), we randomly replace 15% of the tokens with indexed sentinel tokens (like [MASK0]) in the encoder inputs, and require the decoder to recover them via generating a combination of these spans. We follow CodeT5 to employ whole-word masking by sampling spans (span lengths determined by a uniform distribution with a mean of 3) before sub-word tokenization to avoid masking partial words.

**Causal Language Modeling (CLM).** Inspired by Tay et al. (2022); Soltan et al. (2022), we introduce two variants of CLM to optimize our model for auto-regressive generation. In the first variant, we randomly select a pivot location and regard the context before it as the source sequence and the sequence after it as the target output. We denote this variant as a sequence-to-sequence (Seq2Seq) causal LM objective. We restrict the pivot location to be uniformly sampled between 10% and 90% of the whole sequence and prepend a special token [CLM] to the source sequence. The second CLM variant is a decoder-only generation task, where we always pass a [CLM] token to the encoder input and require the decoder to generate the full code

sequence. This task aims to provide more dense supervision signals to train the decoder as an independent full-fledged code generation module.

## 3.2 Bimodal Pretraining on Text-code Data

In the second stage, we pretrain the model using text-code bimodal data at function level (Husain et al., 2019). In this setting, each text-code pair contains a code function and its corresponding docstring describing its semantics. Such a bimodal data format facilitates model training for cross-modal understanding and generation. The bimodal pretraining tasks consist of cross-modal contrastive learning, matching, and causal LM tasks (Fig. 2). See Appendix A for their detailed formulations.

**Text-Code Contrastive Learning.** This task aims to align the feature space of text and code representations by pulling together the representations of positive text-code pairs and pulling apart the negative pairs. Guo et al. (2022) demonstrated the benefits of such learning task for code understanding. This task only activates the encoder, which encodes a text or code snippet into a representation through bidirectional self-attention (Vaswani et al., 2017). Similar to BERT (Devlin et al., 2019), we prepend a special token [CLS] to the input and regard its output embeddings at the final layer as the representations of the corresponding input text or code. We further add a linear layer and use L2 normalization to map the output to $256\text{-}d$ embeddings. To enrich the negative samples, we use a momentum encoder to store embeddings of samples from previous mini-batches, as similarly adopted by (He et al., 2020; Li et al., 2022a). Specifically, the momentum encoder maintains a queuing system that enqueues the samples in the current mini-batch and dequeues the samples in the oldest mini-batch.

**Text-Code Matching.** This task activates the decoder and aims to predict whether a text and code snippet share the same semantics. Such task enables model to learn better bimodal representations that capture the fine-grained alignment between text and code modalities. Given a code sample, the decoder first passes it to an embedding layer and a causal self-attention layer. The representations are then passed to a cross-attention layer which queries relevant signals from the text representations (received from the encoder). A task-specific [Match] token is prepended to the code input sequence to inform the decoder of the text-code matching functionality, and an [EOS] token is appended to the end of the code input. Since the decoder employs causal self-attention masks and only the last decoder token can attend to the whole context, we treat the output embedding of [EOS] at the last layer as the text-code alignment representation. Finally, we use a linear layer on top of the output embedding of the decoder for a binary matching task, predicting whether a text-code pair is positive (matched) or negative (unmatched).

**Text-Code Causal LM.** This task activates both encoder and decoder and focuses on a cross-modal generative objective through a dual multimodal conversion: text-to-code generation and code-to-text generation. Specifically, when the input is a text sample, we prepend a [CDec] token to the input sequence to the decoder. In this case, the decoder operates under code generation functionality. Alternatively, when the input is a code sample, we prepend a [TDec] token to the input sequence to the decoder. The decoder operates under text generation functionality in this case. This type of Causal LM has been shown to be an effective learning objective to close the pretrain-finetune gap for generative downstream tasks (Wang et al., 2021b).

### 3.3 Compute-efficient Pretraining with Frozen Off-the-shelf LLMs

To efficiently scale up the model without the need of pretraining from scratch, we propose a compute-efficient pretraining strategy to initialize model components (i.e. encoder and decoder) of CodeT5+ with off-the-shelf pretrained LLMs (Nijkamp et al., 2023b) (see the rightmost diagram of Fig. 2). For this extension, inspired by (Li et al., 2022b), we employ a "shallow encoder and deep decoder" architecture instead of encoder and decoder of the same size in conventional T5 models (Raffel et al.,

2020; Wang et al., 2021b). As noted by Li et al. (2022b), the decoder is often required to deal with a higher level of complexity in generation tasks and thus, should be enhanced with more parameters.

To connect the separately pretrained encoder and decoder, we insert randomly initialized cross-attention layers to decoder blocks after the self-attention layers. For efficient tuning, we only insert cross-attention layers to the top-$L$ decoder layers ($L$=1 in our experiments). We only keep the small encoder and cross-attention layers trainable while freezing the majority of the decoder parameters. We also explored other advanced designs such as adding a gating function to improve training stability or inserting multiple cross-attention layers at a certain frequency (Alayrac et al., 2022). However, we did not observe significant performance improvement and these design choices would introduce too expensive computation overhead.

### 3.4 Adaptation to Downstream Understanding and Generation Tasks

After the two stages of pretraining, CodeT5+ can flexibly operate in various modes to support different tasks, including Seq2Seq generation tasks, decoder-only tasks, and understanding-based tasks:

**Seq2Seq Generation Tasks.** As an encoder-decoder model, CodeT5+ can be naturally adapted to a variety of Seq2Seq generation tasks such as code generation and summarization. We also adapt CodeT5+ as a retrieval-augmented generation model, using the encoder to retrieve code snippets, which are then used by both the encoder and decoder for code generation.

**Decoder-only Tasks.** In this setting, we always feed a [CLM] token to the encoder input and pass the source sequence to the decoder as the prefix context. We freeze the weights of the encoder and the cross-attention layers in the decoder. This strategy only activates parts of the decoder and reduces about half of the total model parameters. We use next-line code completion tasks to evaluate the decoder-only generation capability of CodeT5+.

**Understanding Tasks.** CodeT5+ can support these understanding tasks in two ways: first, it employs the encoder to obtain text/code embeddings, which can be either passed to a binary classifier for detection tasks; alternatively, the encoder can be combined with the decoder to predict the text-code matching scores for text-to-code retrieval tasks.

## 4 Pretraining and Instruction Tuning

Additional pretraining and finetuning setups can be found in Appendix B, C, and E.

**Pretraining Dataset.** We enlarge the pretraining dataset of CodeSearchNet (Husain et al., 2019) with the recently released GitHub Code dataset[2]. We select nine PLs (Python, Java, Ruby, JavaScript, Go, PHP, C, C++, C#) and filter the dataset by preserving only permissively licensed code[3] and files with 50 to 2000 tokens. Besides, we filter out the overlapped subset with CodeSearchNet and other downstream tasks covered in our evaluation by checking their GitHub repository names. Note that although we employ the deduplicated data version in which duplicates are filtered out based on the exact match, there might be some potential remaining duplicates. However, we do not expect any remaining duplication will impact our model performance significantly. We use the CodeT5 tokenizer to tokenize the multilingual dataset, resulting in 51.5B tokens, ~50x larger than CodeSearchNet.

**Pretraining Setup.** We pretrained two groups of CodeT5+ models: 1) CodeT5+ 220M and 770M that are trained from scratch following T5's architecture (Raffel et al., 2020) (T5-base and large respectively), 2) CodeT5+ 2B, 6B, 16B in which the decoders are initialized from CodeGen-mono 2B, 6B, 16B models (Nijkamp et al., 2023b) and its encoders are initialized from CodeGen-mono 350M. Note that following our model scaling strategy, the latter group of CodeT5+ models introduce insignificant trainable parameters (the 350M encoder plus one cross-attention layer of 36M, 67M, 151M for 2B, 6B, 16B models respectively) compared to the original CodeGen models. We employ the CodeT5 tokenizer and CodeGen tokenizer for these two groups of models respectively. In pretraining, we adopt a stage-wise strategy to pretrain CodeT5+ first on the large-scale unimodal dataset and then on the smaller bimodal dataset on a cluster with 16 A100-40G GPUs on Google Cloud Platform.

**Instruction Tuning.** In the NLP domain, recent work (Wang et al., 2022; Taori et al., 2023) studied the benefits of data augmentation techniques on pretrained LMs with synthetic instruction data. Models finetuned with this type of data can better understand natural language instructions and demonstrate improved alignment with the corresponding tasks (Wang et al., 2022; Ouyang et al., 2022). We are motivated to transfer this technique to the code domain to improve our CodeT5+ models. Following Taori et al. (2023), we employ over 20k instruction data in the code domain curated by Chaudhary (2023). The data is generated by letting pretrained LLMs i.e. text-davinci-003, generate novel tasks, including task instructions, inputs (if any), and expected outputs. We trained our models on this augmented dataset for up to 3 epochs and denote the instruction-tuned models as "InstructCodeT5+". Note that the instruction data are generated fully independently from any downstream evaluation tasks and we still evaluate these models in a zero-shot manner.

## 5 Experiments

We extensively evaluate CodeT5+ on a wide range of code understanding and generation tasks over 20+ code-related datasets across 9 different programming languages (PLs). In addition, we consider a variety of evaluation settings including zero-shot, instruction tuning, task-specific finetuning. Additional results can be found in Appendix D.

**Baselines.** We developed a family of CodeT5+ models, with model sizes ranging from 220M to 16B. We compare CodeT5+ with 3 types of models: encoder-only, decoder-only, and encoder-decoder.

- For **encoder-only** models, we consider RoBERTa (Liu et al., 2019), CodeBERT (Feng et al., 2020), GraphCodeBERT (Guo et al., 2021), SYNCOBERT (Wang et al., 2021a) and UniXcoder (Guo et al., 2022) that incorporates contrastive learning. Note that UniXcoder can be also viewed as decoder-only model as it employs UniLM-style masking (Dong et al., 2019).

- For **decoder-only** models, we consider GPT-2 (Radford et al., 2019) and CodeGPT (Lu et al., 2021), and also consider models of very large scales (up to 540B) such as PaLM (Chowdhery et al., 2022), GPT-4 (OpenAI, 2023), Codex (Chen et al., 2021), LLaMA (Touvron et al., 2023), CodeGen (Nijkamp et al., 2023b), Incoder (Fried et al., 2022), GPT-J (Wang and Komatsuzaki, 2021), GPT-Neo and GPT-NeoX (Black et al., 2022), MIM (Nguyen et al., 2023), CodeGeeX (Zheng et al., 2023). We also compare with Replit (replit, 2023) and StarCoder (Li et al., 2023) which are concurrent work with ours.

---

[2]https://huggingface.co/datasets/codeparrot/github-code

[3]Permissive licenses: "mit", "apache-2", "bsd-3-clause", "bsd-2-clause", "cc0-1.0", "unlicense", "isc"

Table 1: Results of *pass@k(%)* on HumanEval.

| Model | Model size | pass@1 | pass@10 | pass@100 |
|---|---|---|---|---|
| Closed-source models | | | | |
| LaMDA | 137B | 14.0 | - | 47.3 |
| AlphaCode | 1.1B | 17.1 | 28.2 | 45.3 |
| MIM | 2.7B | 30.7 | 48.2 | 69.6 |
| PaLM | 62B | 15.9 | - | 46.3 |
| PaLM | 540B | 26.2 | - | 76.2 |
| code-cushman-001 | - | 33.5 | 54.3 | 77.4 |
| code-davinci-002 | - | 47.0 | **74.9** | **92.1** |
| GPT-3.5 | - | 48.1 | - | - |
| GPT-4 | - | **67.0** | - | - |
| Open-source models | | | | |
| GPT-J | 6B | 11.6 | 15.7 | 27.7 |
| InCoder | 6B | 15.2 | 27.8 | 47.0 |
| GPT-NeoX | 20B | 15.4 | 25.6 | 41.2 |
| CodeGeeX | 13B | 22.9 | 39.6 | 60.9 |
| LLaMA | 13B | 15.8 | - | 52.5 |
| LLaMA | 65B | 23.7 | - | **79.3** |
| Replit | 3B | 21.9 | - | - |
| StarCoder | 15B | 33.6 | - | - |
| CodeGen-mono | 2B | 23.7 | 36.6 | 57.0 |
| CodeGen-mono | 6B | 26.1 | 42.3 | 65.8 |
| CodeGen-mono | 16B | 29.3 | 49.9 | 75.0 |
| CodeT5+ | 220M | 12.0 | 20.7 | 31.6 |
| CodeT5+ | 770M | 15.5 | 27.2 | 42.7 |
| CodeT5+ | 2B | 24.2 | 38.2 | 57.8 |
| CodeT5+ | 6B | 28.0 | 47.2 | 69.8 |
| CodeT5+ | 16B | 30.9 | 51.6 | 76.7 |
| InstructCodeT5+ | 16B | **35.0** | **54.5** | 77.9 |
| Open-source models + generation strategies | | | | |
| StarCoder (prompted) | 15B | 40.8 | - | - |
| CodeGen-mono w/ CodeT | 16B | 36.7 | 59.3 | - |
| CodeT5+ w/ CodeT | 16B | 38.5 | 63.6 | 77.1 |
| InstructCodeT5+ w/ CodeT | 16B | **42.9** | **67.8** | **78.7** |

Table 2: Results of *pass@k(%)* on math programming.

| Model | Model size | MathQA-Python pass@80 | GSM8K-Python pass@100 |
|---|---|---|---|
| Few-shot learning results | | | |
| code-davinci | - | **42.0** | 71.0 |
| LLaMA | 33B | - | 53.1 |
| LLaMA | 65B | - | 69.7 |
| Minerva | 62B | - | 68.5 |
| Minerva | 540B | - | **78.5** |
| Finetuning results | | | |
| LaMDA | 137B | 81.2 | - |
| GPT-Neo | 125M | 84.7 | - |
| GPT-Neo | 2.7B | - | 41.4 |
| CodeGen-mono | 350M | 83.1 | 38.7 |
| CodeGen-mono | 2B | 85.6 | 47.8 |
| CodeT5 | 220M | 71.5 | 58.4 |
| CodeT5+ | 220M | 85.6 | 70.5 |
| CodeT5+ | 770M | **87.4** | **73.8** |

• For **encoder-decoder**, we use PLBART (Ahmad et al., 2021) and CodeT5 (Wang et al., 2021b).

Note that billion-parameter LLMs such as Codex and CodeGen typically use most of the source code from GitHub for model training and do not remove any overlap with the downstream tasks covered in this work as we did. Therefore, it is difficult to ensure a fair comparison with these models in those tasks, especially the code completion tasks. Moreover, these models are very expensive to perform task-specific finetuning, and hence, they are often employed only on the zero-shot evaluation. In this work, we mainly compare CodeT5+ with these LLMs in the zero-shot HumanEval code generation task (Sec. 5.1). In other experiments, we focus on the finetuning setting and compare our models with smaller-scale LMs.

## 5.1 Zero-shot Code Generation Evaluation

We first evaluate the zero-shot code generation capabilities of our model on the HumanEval benchmark (Chen et al., 2021), where we activate both encoder and decoder modules from CodeT5+. In this experiment, we follow Nijkamp et al. (2023b) to continue to pretrain our CodeT5+ models on the Python subset for another epoch using causal LM objective to adapt them for Python code generation. We evaluate the model performance by

testing generated codes against unit tests and report the passing rate *pass@k* ($k = \{1, 10, 100\}$).

As shown in Table 1, our instruction-tuned CodeT5+ ("InstructCodeT5+") 16B can improve the performance against other open code LLMs, achieving new SoTA of 35.0% pass@1 and 54.5% pass@10. Particularly, as an open model, it even outperforms the OpenAI code-cushman-001 model across all metrics. We also observed that our small-sized models of 220M and 770M already match or outperform much larger code LLMs, e.g., CodeT5+ 770M's 15.5% pass@1 compared to Incoder 6B's 15.2%, GPT-NeoX 20B's 15.4%, and PaLM 62B's 15.9%. Besides, we observed that compared to the CodeGen models of similar sizes (Nijkamp et al., 2023b), CodeT5+ obtains consistent performance gains from 2B to 16B models. These superior results against decoder-only baselines demonstrate the advantage of the encoder-decoder architecture of CodeT5+ and validate the effectiveness of our proposed compute-efficient pretraining strategy. We also evaluated the models with enhancement strategies following CodeT Chen et al. (2023). We find that this strategy can select better code candidates and bring the performance gains, achieving up to 42.9% pass@1 and 67.8% pass@10.

## 5.2 Evaluation on Math Programming

We consider two math programming benchmarks MathQA-Python (Austin et al., 2021) and GSM8K (Cobbe et al., 2021). The task is to generate Python programs to solve mathematical problems described in texts, where code correctness is measured based on the execution outputs of the generated programs (pass@k). We compare our models with very large decoder-only LMs such as Minerva (Lewkowycz et al., 2022) that is ini-

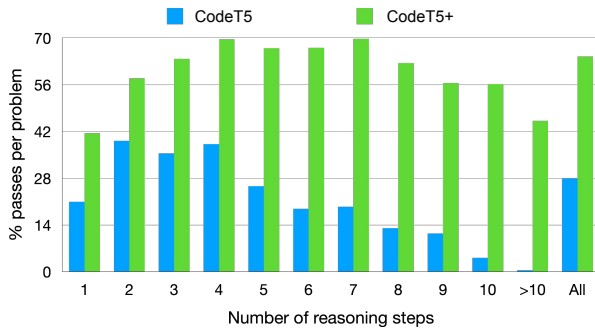

Figure 4: Results of MathQA-Python by problem complexity (i.e. the number of reasoning steps required).

Table 3: Results on line-level code completion.

| Model | PY150 | | JavaCorpus | |
|---|---|---|---|---|
| | EM | Edit Sim | EM | Edit Sim |
| CodeGPT 124M | 42.37 | 71.59 | 30.60 | 63.45 |
| UniXcoder 125M | 43.12 | 72.00 | 32.90 | 65.78 |
| CodeGen-multi 350M | 42.47 | 70.67 | 35.47 | 69.22 |
| PLBART 140M | 38.01 | 68.46 | 26.97 | 61.59 |
| CodeT5 220M | 36.97 | 67.12 | 24.80 | 58.31 |
| CodeT5+ 220M | 43.42 | 73.69 | 35.17 | 69.48 |
| CodeT5+ 770M | **44.86** | **74.22** | **37.90** | **72.25** |

Table 4: Results of MRR on Text-to-Code Retrieval.

| Model | CodeSearchNet | | | | | | | CosQA | AdvTest |
|---|---|---|---|---|---|---|---|---|---|
| | Ruby | JS | Go | Python | Java | PHP | Overall | | |
| CodeBERT 125M | 67.9 | 62.0 | 88.2 | 67.2 | 67.6 | 62.8 | 69.3 | 65.7 | 27.2 |
| GraphCodeBERT 125M | 70.3 | 64.4 | 89.7 | 69.2 | 69.1 | 64.9 | 71.3 | 68.4 | 35.2 |
| SYNCOBERT 125M | 72.2 | 67.7 | 91.3 | 72.4 | 72.3 | 67.8 | 74.0 | - | 38.3 |
| UniXcoder 125M | 74.0 | 68.4 | 91.5 | 72.0 | 72.6 | 67.6 | 74.4 | 70.1 | 41.3 |
| CodeGen-multi 350M | 66.0 | 62.2 | 90.0 | 68.6 | 70.1 | 63.9 | 70.1 | 64.8 | 34.8 |
| PLBART 140M | 67.5 | 61.6 | 88.7 | 66.3 | 66.3 | 61.1 | 68.6 | 65.0 | 34.7 |
| CodeT5 220M | 71.9 | 65.5 | 88.8 | 69.8 | 68.6 | 64.5 | 71.5 | 67.8 | 39.3 |
| CodeT5+ 220M | 77.7 | 70.8 | 92.4 | 75.6 | 76.1 | 69.8 | 77.1 | 72.7 | 43.3 |
| CodeT5+ 770M | **78.0** | **71.3** | **92.7** | **75.8** | **76.2** | **70.1** | **77.4** | **74.0** | **44.7** |

tialized with pretrained PaLM (Chowdhery et al., 2022) and further finetuned with large-scale scientific corpora. Note that some of the baselines are enhanced with generation strategies, such as GPT-Neo using self-sampling optimization (Ni et al., 2022), and LLaMA and Minerva using majority voting (Lewkowycz et al., 2022).

Table 2 shows that CodeT5+ achieves significant performance gains, outperforming many code LLMs of much larger sizes. Specifically, our CodeT5+ 770M achieves new SoTA results of 87.4 pass@80 on MathQA-Python and very competitive results of 73.8 pass@100 on GSM8K-Python. On GSM8K-Python, CodeT5+ 770M achieves the best finetuning results against other larger models (e.g., LaMDA 137B and GPT-Neo 2.7B), and outperforms LLaMA 65B and Minerva 62B in the few-shot evaluation setting. In Fig. 4, we further analyze the model performance of CodeT5+ by the problem complexity on MathQA-Python compared to CodeT5. For each problem, we extract the number of reasoning steps required to solve the problem. We observe that CodeT5+ is more robust against the complexity of the problems compared to CodeT5, where CodeT5 model performance tends to deteriorate drastically as the number of reasoning steps increases. In CodeT5+, the downward trend is a lot less severe and the model still achieves good results in very complex tasks (more than 10 steps).

### 5.3 Evaluation on Code Completion

We evaluate the decoder-only generation capability of CodeT5+ through a line-level code completion task, which aims to complete the next code line based on the previous code contexts. We employ PY150 (Raychev et al., 2016) and JavaCorpus (Allamanis and Sutton, 2013) from CodeXGLUE, and use exact match (EM) accuracy and Levenshtein edit similarity (Svyatkovskiy et al., 2020) as the

metrics. In this task, we employ a decoder-only model from CodeT5+ so that only about half of the total model parameters are activated.

Table 3 shows that both CodeT5+ (in decoder-only mode) and decoder-only models (the top block) significantly outperform encoder-decoder models (the middle block), validating that decoder-only models can better suit the code completion task in nature. Specifically, CodeT5+ 220M already surpasses UniXcoder and is comparable to CodeGen-multi 350M, while the 770M one further sets new SoTA results in both metrics. In particular, CodeT5+ 220M yields substantial improvements over CodeT5 220M by +6.5 EM and +10.4 EM scores on PY150 and JavaCorpus respectively. This is mainly due to our causal LM objectives that allows the decoder to see longer sequences and thus have a better causal generation capability.

### 5.4 Evaluation on Text-to-Code Retrieval

We evaluate the code understanding capabilities of CodeT5+ through text-to-code retrieval tasks across multiple PLs. This task aims to find the most semantically related code snippet at the function level from a collection of candidate codes based on a natural language query. We consider three datasets for evaluation: CodeSearchNet (Husain et al., 2019), CosQA (Huang et al., 2021), and AdvTest (Lu et al., 2021), which are curated from the original CodeSearchNet by filtering data with low-quality queries, adopting real-world queries from a modern search engine, and obfuscating identifiers to normalize the code. In this task, we activate both encoder and decoder of CodeT5+ and use Mean Reciprocal Rank (MRR) as the evaluation metric.

Table 5: Ablation results of CodeT5+: a) no causal LM objective during stage-1 pretraining, b) no matching or causal LM objective during stage-2 pretraining.

| Model | Code Completion | | Math Programming | |
|---|---|---|---|---|
| | PY150 | JavaCorpus | MathQA-PY | GSM8K-PY |
| | EM | EM | pass@80 | pass@100 |
| CodeT5+ 770M | **44.9** | **37.9** | **87.4** | **73.8** |
| a) no causal LM | 36.2 | 24.8 | 72.3 | 61.4 |

| Model | Text-to-code Retrieval | | | | | | |
|---|---|---|---|---|---|---|---|
| | Ruby | JS | Go | Python | Java | PHP | Overall |
| CodeT5+ 770M | **78.0** | **71.3** | **92.7** | **75.8** | **76.2** | **70.1** | **77.4** |
| b) no matching | 76.2 | 68.5 | 91.2 | 72.8 | 73.6 | 66.3 | 74.8 |
| no causal LM | 77.3 | 70.6 | 92.4 | 75.7 | 75.6 | 68.9 | 76.8 |

Table 6: Results of retrieval-augmented code generation. EM: Exact Match, B4: BLEU-4, CB: CodeBLEU.

| Model | Java | | | Python | | |
|---|---|---|---|---|---|---|
| | EM | B4 | CB | EM | B4 | CB |
| Retrieval-based | | | | | | |
| BM25 | 0.00 | 4.90 | 16.00 | 0.00 | 6.63 | 13.49 |
| SCODE-R 125M | 0.00 | 25.34 | 26.68 | 0.00 | 22.75 | 23.92 |
| CodeT5+ 220M | 0.00 | 28.74 | 31.00 | 0.00 | 27.30 | 26.51 |
| Generative | | | | | | |
| CodeBERT 125M | 0.00 | 8.38 | 14.52 | 0.00 | 4.06 | 10.42 |
| GraphCodeBERT 125M | 0.00 | 7.86 | 14.53 | 0.00 | 3.97 | 10.55 |
| PLBART 140M | 0.00 | 10.10 | 14.96 | 0.00 | 4.89 | 12.01 |
| CodeT5+ 220M | 0.00 | 10.33 | 20.54 | 0.00 | 4.40 | 13.88 |
| Retrieval-Augmented Generative | | | | | | |
| REDCODER-EXT 125M+140M | 10.21 | 28.98 | 33.18 | 9.61 | 24.43 | 30.21 |
| CodeT5+ 220M | **11.66** | **33.83** | **40.60** | **11.83** | **31.14** | **36.39** |

From Table 4, our CodeT5+ 220M significantly outperforms all existing encoder-only/decoder-only (the top block) and encoder-decoder models (the middle block). Our CodeT5+ 770M further sets new SoTA results, surpassing the previous SoTA UniXcoder by more than 3 MRR points on all 3 tasks across 8 datasets. This implies CodeT5+ is a robust code retriever to handle queries with diverse formats and PLs. Besides, CodeT5+ 220M yields substantial performance gains over CodeT5 220M, which can be attributed to the text-code contrastive learning and matching objectives that facilitate better unimodal and bimodal representation learning.

## 5.5 Ablation Study

We conduct an ablation study to analyze the impacts of our proposed pretraining objectives: a) casual LM objectives at stage-1 unimodal pretraining on two generative tasks including code completion and math programming, b) text-code matching and causal LM objectives at stage-2 bimodal pretraining on an understanding task of text-to-code retrieval. We employ CodeT5+ 770M and report the results of three representative tasks over 10 datasets in Table 5. In CodeT5+, we found that causal LM objective plays a crucial role in code completion and math programming tasks, observed by a significant performance drop after removing it. This indicates causal LM can complement the span denoising objective and improve the generation capability of our models. Additionally, we found that the text-code matching objective is critical to the retrieval performance (a drop of 2.6 avg. MRR over 6 datasets without it), implying this objective can learn a better bimodal representation that captures the fine-grained alignment between text and code. Besides, we found that retrieval tasks can also benefit from the joint training with causal LM objective despite their task differences.

## 5.6 Unified Retrieval-Augmented Generation

As our model is capable of both code retrieval and generation, it can be naturally exploited as a unified retrieval-augmented generator. To explore this adaptation, we follow Parvez et al. (2021) to evaluate two code generation tasks on Java and Python. We evaluate our models in 3 settings: retrieval-based, generative, and retrieval-augmented (RA) generative. For the retrieval-based setting, we activate our encoder to retrieve the top-1 code sample as the prediction given a text query, while for the RA generative setting, we append the combination of top-$k$ retrieved samples ($k$=1 in our work) to the encoder input and activate the decoder. As shown in Table 6, we found that our CodeT5+ achieves better results in all categories, especially in the retrieval-based and RA generative setting. While the previous SoTA model REDCODER-EXT (Parvez et al., 2021) separately employs GraphCodeBERT as the retriever and PLBART as the generator, our model can be seamlessly deployed as a unified end-to-end system with both retrieval and generation capabilities.

## 6 Conclusion

We propose CodeT5+, a new family of open code LLMs with a dynamic architecture that can flexibly operate in different modes (encoder-only, decoder-only, and encoder-decoder) to support a wide range of code understanding and generation tasks. To train CodeT5+, we introduce a mixture of pretraining tasks to learn rich representations from both unimodal code data and bimodal code-text data. Additionally, it achieves efficient model scaling and better task generalization through integration with frozen LLMs and instruction tuning. Extensive experiments on over 20 code intelligence benchmarks have verified the superiority of our models.

## Limitations

As a family of Transformer LLMs, CodeT5+ requires sufficient pretraining/finetuning data to be able to learn meaningful contextual representations from code. While we could curate these data from public domains such as GitHub, thorough data filtering and preprocessing steps are needed to obtain a good level of data quality for pretraining. During instruction finetuning, a well designed pipeline is needed to obtain high quality instruction-following data, either through manual annotation effort or synthetic data augmentation from other LLMs (e.g. OpenAI GPT models). Moreover, the level of the diversity and quality of data needed to train these types of models is still an open question. Recent attempts such as (Zhou et al., 2023) have highlighted the importance of data quality vs. data scale to efficiently train LLMs while keeping the cost of handling data affordable.

Another limitation of CodeT5+ is the requirement of large GPU resources. With model sizes up to billion parameters, to handle these models efficiently requires access to GPUs during either training and inference time. Specifically, we found that fitting a 16B model into a single A100 GPU requires additional model serving/loading techniques to keep the system memory consumption acceptable. While GPU resources have become more and more accessible to the wider community of practitioners, the cost of training/testing LLMs on large-scale data can accumulate and become too expensive to many individuals.

## Ethics Statement

Advancements in code understanding and generation systems hold immense potential to create positive societal impacts by improving programming accessibility and enhancing developer productivity through natural language interfaces. However, deploying such systems at scale requires careful consideration of various ethical aspects, as extensively discussed by Chen et al. (2021).

One critical concern is the potential risk of generated code summaries or comments incorporating toxic or insensitive language, which can have detrimental effects. Several studies have explored techniques to address this issue, such as reinforcement learning (Ouyang et al., 2022), weighted decoding (Krause et al., 2021) , and safety-specific control tokens (Xu et al., 2020). These approaches aim to ensure non-toxic natural language generation,

promoting responsible and ethical use of large language models for code.

Additionally, it is essential to recognize the broader intellectual property implications of code generation and retrieval systems before deployment. Deep learning models generating code may inadvertently introduce security vulnerabilities. To mitigate this risk, it is crucial to conduct expert reviews and rigorous security assessments before adopting such code. This review process ensures that the generated code meets necessary security standards, safeguarding against potential exploits and vulnerabilities. In code retrieval scenarios, providing appropriate attribution to the source along with the retrieved results is paramount. This attribution not only respects the rights of code authors but also enhances transparency, traceability, and collaboration within the programming community. By acknowledging the original authors and promoting a collaborative, ethical, and legally compliant environment, code retrieval systems can foster knowledge sharing and contribute to a reputable programming ecosystem.

By considering these ethical considerations, we can promote the responsible deployment of large language models for code, maximizing their potential benefits while mitigating potential harms to individuals, communities, and the overall software ecosystem. It is imperative to prioritize safety, non-toxicity, intellectual property rights, security, and collaboration in the development and deployment of these systems, ensuring they align with ethical principles and societal needs.

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

## A  Bimodal Pretraining Details

To expose the model on more diverse set of pretraining data, we employ a stage-wise pretraining process to first train CodeT5+ on large-scale code-only data with span denoising and causal language modeling (CLM) tasks, then train on smaller set of text-code bimodel data using text-code contrastive learning, matching, and causal LM tasks. Below, we provide detailed formulas for text-code contrastive learning and matching tasks at the second-stage pretraining on text-code pairs.

**Text-Code Contrastive Learning**  activates the encoder to learn better unimodal (text/code) representations by computing a similarity score such that parallel text-code pairs have higher scores. Given a text T and a code C, we first learn representations $\mathbf{h}^t$ for text $T$ and $\mathbf{h}^c$ for code $C$ by mapping the [CLS] embeddings to normalized lower-dimensional (256-d) representations from the encoder. Given a batch of $N$ text-code pairs, we obtain text vectors $\{\mathbf{h}^t\}_{i=1}^N$ and code vectors $\{\mathbf{h}^c\}_{i=1}^N$ to compute text-to-code and code-to-text and similarities:

$$s_{i,j}^{t2c} = \mathbf{h}_i^{t\top}\mathbf{h}_j^c, s_{i,j}^{c2t} = \mathbf{h}_i^{c\top}\mathbf{h}_j^t \qquad (1)$$

$$p_i^{t2c}(T) = \frac{\exp\left(s_{i,i}^{t2c}/\tau\right)}{\sum_{j=1}^N \exp\left(s_{i,j}^{t2c}/\tau\right)}, \qquad (2)$$

Table 7: Data statistics of both unimodal and bimodal (CodeSearchNet) pretraining data.

| Dataset | Language | # Sample | Total size |
|---|---|---|---|
| Ours | Ruby | 2,119,741 | 37,274,876 files |
| | JavaScript | 5,856,984 | |
| | Go | 1,501,673 | |
| | Python | 3,418,376 | |
| | Java | 10,851,759 | |
| | PHP | 4,386,876 | |
| | C | 4,187,467 | |
| | C++ | 2,951,945 | |
| | C# | 4,119,796 | |
| CSN | Ruby | 49,009 | 1,929,817 text-code pairs at function level |
| | JavaScript | 125,166 | |
| | Go | 319,132 | |
| | Python | 453,772 | |
| | Java | 457,381 | |
| | PHP | 525,357 | |

$$p_i^{c2t}(C) = \frac{\exp\left(s_{i,i}^{c2t}/\tau\right)}{\sum_{j=1}^N \exp\left(s_{i,j}^{c2t}/\tau\right)} \qquad (3)$$

where $s_{i,j}^{t2c}$ represents text-to-code similarity of text of $i$-th pair and code of $j$-th pair, and $s_{i,j}^{c2t}$ is the code-to-text similarity, $\tau$ is learned temperature parameter. $p_i^{t2c}(T)$ and $p_i^{c2t}(C)$ are the softmax-normalized text-to-code and code-to-text similarities for the $i$-th text and code.

Let $\mathbf{y}^{t2c}(T)$ and $\mathbf{y}^{c2t}(C)$ denote the ground-truth one-hot similarity, where negative pairs have a probability of 0 and the positive pair has a probability of 1. The text-code contrastive loss from a corpus $D$ of text-code pairs is defined as the cross-entropy H between $\mathbf{p}$ and $\mathbf{y}$:

$$\mathcal{L}_{tcc} = \frac{1}{2}\mathbb{E}_{(T,C)\sim D}[H(\mathbf{y}^{t2c}(T), \mathbf{p}^{t2c}(T)) + H(\mathbf{y}^{c2t}(C), \mathbf{p}^{c2t}(C))] \quad (4)$$

**Text-Code Matching**  activates the decoder with the bimodal matching functionality to predict whether a pair of text and code is positive (matched) or negative (unmatched). We employ the output embedding of the [EOS] token as the fused bimodal representation for a text-code pair $(T, C)$, as this token attends to all the previous context for the text-code pair input. Followed by a linear layer and softmax, we compute a two-class probability $p^{tcm}(T, C)$ and define the matching loss:

$$\mathcal{L}_{tcm} = \mathbb{E}_{(T,C)\sim D}[H(\mathbf{y}^{tcm}(T, C), \mathbf{p}^{tcm}(T, C))] \quad (5)$$

where $\mathbf{y}^{tcm}(T, C)$ is a 2-dimensional one-hot vector representing the ground-truth label.

Figure 5: **Example generated instruction data:** we demonstrate some examples of instruction data used to finetune CodeT5+ to better align our models to natural language instructions. The instruction corpus contains novel tasks, such as text-to-SQL generation and Python code optimization.

**Text-Code Causal LM.** This task focuses on a cross-modal causal LM objective between text and code through a dual multimodal conversion: text-to-code generation and code-to-text generation (i.e. code summarization). Let $\mathcal{L}_{t2c}$ and $\mathcal{L}_{c2t}$ denote the losses for text-to-code and code-to-text generation. The full second-stage pretraining loss of our CodeT5+ is:

$$\mathcal{L} = \mathcal{L}_{tcc} + \mathcal{L}_{tcm} + \mathcal{L}_{t2c} + \mathcal{L}_{c2t} \qquad (6)$$

## B  Pretraining

### B.1  Pretraining Dataset

We report the data statistics of both unimodal code and bimodal text-code pretraining datasets in Table 7. From the table, we can see that our curated dataset from GitHub code has a much larger data size at the file level than the CodeSearchNet bimodal data at the function level, allowing our model to learn rich representations in the first stage of pretraining. Different from CodeT5 (Wang et al., 2021b) which employs both unimodal and bimodal data in CodeSearchNet (Husain et al., 2019), we only employ its bimodal subset for the second stage pretraining of our CodeT5+. We use this stage to mainly adapt our model to text-code related tasks like text-to-code retrieval and generation.

### B.2  Pretraining Setup

In pretraining, we adopt a stage-wise strategy to pretrain CodeT5+ first on the large-scale unimodal dataset and then on the smaller bimodal dataset. In the first stage, we warm up the model with the span denoising task for $10k$ training steps, and then joint

training with the two CLM tasks with equal weights for $100k$ steps. We employ a linear decay learning rate (LR) scheduler with a peak learning rate of 2e-4 and set the batch size to 2048 for denoising and 512 for CLM. To prepare the input and output data, we set the maximum length to 512 for the denoising task, and set the maximum lengths to 768 and 600 for source and target sequences for the code completion CLM, 1 and 1024 for the decoder-only generation CLM. In the second stage, we jointly optimize four losses of contrastive learning, matching, and two CLM losses with equal weights for 10 epochs with a batch size of 256. We employ a peak learning rate of 1e-4 and set the maximum sequence lengths to 420 and 128 for code and text.

In all experiments, we employ an AdamW optimizer (Loshchilov and Hutter, 2019) with a 0.1 weight decay. We also employ the DeepSpeed's ZeRO Stage 2 (Rasley et al., 2020) with mixed precision training of FP16 for training acceleration. For the training of CodeT5+ 2B, 6B, and 16B, we use FP16 frozen decoder weights and keep other trainable weights in FP32. We use DeepSpeed ZeRO Stage 3's parameter partition for CodeT5+ 6B and 16B models.

## C  Instruction Tuning

Fig. 5 illustrates some examples of the generated instruction data. Note that as we rely on LM-generated data, including the annotations of expected outputs, not all of the data is perfectly correct. For instance, the example of the code optimization task in Fig. 5 contains a wrong output. Wang et al. (2022) treated these examples as data

Table 8: Results of BLEU-4 on code summarization.

| Model | Ruby | JS | Go | Python | Java | PHP | Overall |
|---|---|---|---|---|---|---|---|
| RoBERTa 125M | 11.17 | 11.90 | 17.72 | 18.14 | 16.47 | 24.02 | 16.57 |
| CodeBERT 125M | 12.16 | 14.90 | 18.07 | 19.06 | 17.65 | 25.16 | 17.83 |
| UniXcoder 125M | 14.87 | 15.85 | 19.07 | 19.13 | 20.31 | 26.54 | 19.30 |
| CodeGen-multi 350M | 13.48 | 16.54 | 18.09 | 18.31 | 19.41 | 24.41 | 18.37 |
| PLBART 140M | 14.11 | 15.56 | 18.91 | 19.30 | 18.45 | 23.58 | 18.32 |
| CodeT5 220M | 15.24 | 16.16 | 19.56 | 20.01 | 20.31 | 26.03 | 19.55 |
| CodeT5+ 220M | 15.51 | 16.27 | 19.60 | 20.16 | 20.53 | 26.78 | 19.81 |
| CodeT5+ 770M | 15.63 | 17.93 | 19.64 | 20.47 | 20.83 | 26.39 | **20.15** |

noise and the tuned models still benefit from the majority of the synthetic instruction dataset.

## D  Additional Experimental Results

In this section, we provide additional experimental results including code summarization (Appendix D.1), two understanding tasks of code defect detection and clone detection from the CodeXGLUE (Lu et al., 2021) (Appendix D.2), more analysis on retrieval-augmented code generation (Appendix D.3), and more qualitative results in math programming tasks (Appendix D.4).

### D.1  Code Summarization from CodeXGLUE

The code summarization task aims to summarize a code snippet into natural language docstrings. We employ the clean version of CodeSearchNet dataset (Husain et al., 2019) in six programming languages to evaluate our models for this task. We employ BLEU-4 (Lin and Och, 2004) as the performance metric which measures the token-based similarity between predicted and ground-truth summaries. From pretrained CodeT5+, we activate both encoder and decoder for this task.

From Table 8, we found that encoder-decoder models (CodeT5 and CodeT5+) generally outperform both encoder-only models (Feng et al., 2020) and decoder-only models (Nijkamp et al., 2023b), as well as the UniLM-style model UniXcoder (Guo et al., 2022). This observation demonstrates the benefit of using the encoder-decoder architecture in CodeT5+ to better encode code contexts and generate more accurate code summaries. Finally, we also observed some performance gains against CodeT5 (Wang et al., 2021b), indicating the advantage of our proposed mixture of diverse pretraining learning objectives in addition to the span denoising objective in CodeT5.

### D.2  Code Defect Detection and Clone Detection from CodeXGLUE

Defect detection is to predict whether a code is vulnerable to software systems or not, while clone

Table 9: Results on two understanding tasks: code defect detection and code clone detection.

| Model | Defect | Clone Detection | | |
|---|---|---|---|---|
| | Acc | Rec | Prec | F1 |
| CodeBERT 125M | 62.1 | 94.7 | 93.4 | 94.1 |
| GraphCodeBERT 125M | - | 94.8 | 95.2 | 95.0 |
| UniXcoder 125M | - | 92.9 | **97.6** | **95.2** |
| CodeGen-multi 350M | 63.1 | 94.1 | 93.2 | 93.6 |
| PLBART 140M | 63.2 | 94.8 | 92.5 | 93.6 |
| CodeT5 220M | 65.8 | 95.1 | 94.9 | 95.0 |
| CodeT5+ 220M | 66.1 | 96.4 | 94.1 | **95.2** |
| CodeT5+ 770M | **66.7** | **96.7** | 93.5 | 95.1 |

detection aims to measure the similarity between two code snippets and predict whether they have a common functionality. We use benchmarks from CodeXGLUE (Lu et al., 2021) and use accuracy and F1 score as the metrics. In Table 9, we can see CodeT5+ models achieve new SoTA accuracy of 66.7% on the defect detection task. For the clone detection task, our model achieves comparable results to SoTA models, where the performance increase tends to be saturated, observed by the close performance gaps between multiple baselines.

### D.3  More Analysis on Retrieval-augmented Code Generation

We conduct an ablation study to analyze the effects of top-$k$ retrievals in retrieval-augmented code generation tasks and report the results in Table 10 . We found that increasing the number of retrievals can boost model performance which becomes saturated when $k$=5. This saturation is due to the maximum sequence length of 600, which might not be able to accommodate a large number of retrieved code samples. Overall, our CodeT5+ significantly outperforms the prior SOTA baseline which uses top-10 retrievals in all cases, even with only a top-1 retrieved code.

We further include a qualitative case in Fig. 6, where we found that the retrieved code provides crucial contexts (e.g., use "urllib3" for an HTTP request) to guide the generative process for more correct prediction. In contrast, the generative-only model gives an incorrect prediction that only captures the concepts of "download" and "compress".

### D.4  Qualitative Results in Math Programming tasks

For math programming tasks, we provide qualitative examples predicted by our models in Fig. 7 and Fig. 8. Overall, we found CodeT5+ is able to

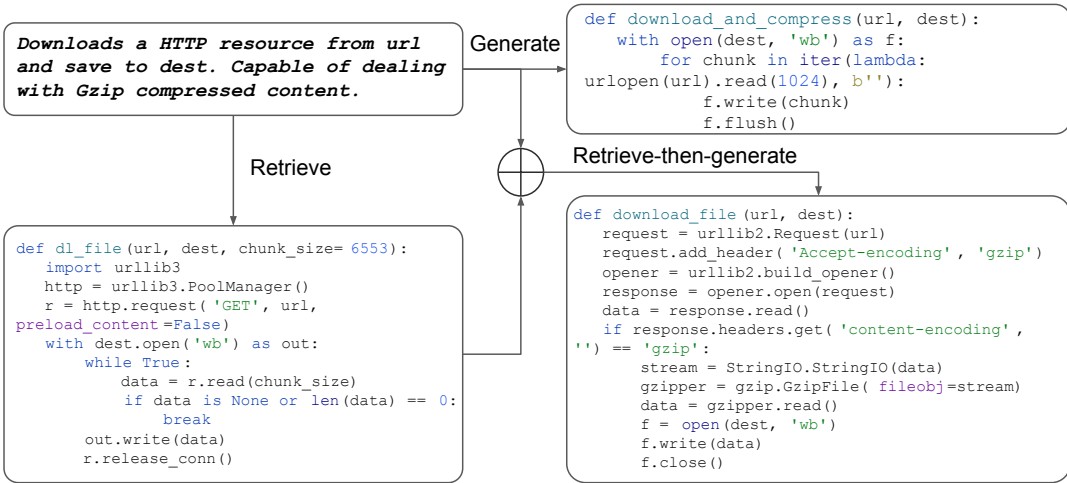

```
Downloads a HTTP resource from url
and save to dest. Capable of dealing
with Gzip compressed content.
```

Generate

```python
def download_and_compress(url, dest):
    with open(dest, 'wb') as f:
        for chunk in iter(lambda:
urlopen(url).read(1024), b''):
            f.write(chunk)
            f.flush()
```

Retrieve

Retrieve-then-generate

```python
def dl_file(url, dest, chunk_size= 6553):
    import urllib3
    http = urllib3.PoolManager()
    r = http.request('GET', url,
preload_content=False)
    with dest.open('wb') as out:
        while True:
            data = r.read(chunk_size)
            if data is None or len(data) == 0:
                break
            out.write(data)
        r.release_conn()
```

```python
def download_file(url, dest):
    request = urllib2.Request(url)
    request.add_header('Accept-encoding', 'gzip')
    opener = urllib2.build_opener()
    response = opener.open(request)
    data = response.read()
    if response.headers.get('content-encoding',
'') == 'gzip':
        stream = StringIO.StringIO(data)
        gzipper = gzip.GzipFile( fileobj=stream)
        data = gzipper.read()
    f = open(dest, 'wb')
    f.write(data)
    f.close()
```

Figure 6: **Example code generation output:** Our CodeT5+ retrieval-augmented generation model could retrieve relevant code context and use it to facilitate better code generation.

Table 10: Effects of varying top-$k$ retrievals in retrieval-augmented code generation tasks with our CodeT5+ 220M compared to the prior SOTA model of REDCODER-EXT that employs top-10 retrievals. EM: Exact Match, B4: BLEU-4, CB: CodeBLEU.

| Model | Java | | | Python | | |
|---|---|---|---|---|---|---|
| | EM | B4 | CB | EM | B4 | CB |
| SOTA (top-10) | 10.21 | 28.98 | 33.18 | 9.61 | 24.43 | 30.21 |
| Ours | | | | | | |
| top-1 | 11.66 | **33.83** | 40.60 | 11.83 | 31.14 | 36.39 |
| top-2 | 11.57 | 33.26 | 40.74 | 11.78 | **31.21** | 36.58 |
| top-3 | 12.29 | 33.10 | 41.71 | 12.48 | 30.92 | 37.31 |
| top-4 | 12.42 | 32.08 | 41.94 | 12.73 | 30.40 | 37.60 |
| top-5 | **13.02** | 32.42 | **42.28** | **12.93** | 30.52 | **37.87** |
| top-10 | 12.86 | 31.38 | 42.24 | 12.84 | 29.79 | 37.79 |

generate decent programs that can solve the math problems in various levels of difficulties, i.e. from simple math operations to more complex problems with multiple reasoning steps. From the rightmost example of Fig. 8, we found that CodeT5+ is able to leverage some external libraries such as *math* when synthesizing the solutions.

## E    Finetuning on Downstream Tasks

### E.1    Text-to-Code Retrieval

Text-to-code retrieval (or code search), is the task of finding the best code sample that is most relevant to a natural language query, from a collection of code candidates. We experiment CodeT5+ with three major benchmarks: CodeSearchNet (CSN) (Husain et al., 2019), CosQA (Huang et al., 2021), and AdvTest (Lu et al., 2021). CSN consists of six programming languages in total, and the dataset is curated by filtering low-quality queries through handcrafted rules, following (Guo et al., 2021). For

instance, an example handcraft rule is to filter examples in which the number of tokens in query is shorter than 3 or more than 256.

CosQA and AdvTest are two related benchmarks that are both derived from the CSN data. Specifically, instead of natural language queries, CosQA uses logs from Microsoft Bing search engine as queries, each of which is annotated by 3 human annotators (Huang et al., 2021). AdvTest is created from the Python split of the CSN data but the code samples are normalized with obfuscated variable names to better evaluate the understanding abilities of current models. For training, we set the maximum sequence to 350 and 64 for code and text. We set the learning rate as 2e-5 and finetune the model for 10 epochs. We employ distributed training on 8 A100s and the total batch size is 64. For momentum encoders, we maintain a separate text/code queue with a size of 57600, and allow the matching decoder to retrieve 64 hard negatives from the queues for hard negative mining.

### E.2    Code Summarization

Code summarization is the task of generating a natural language summary of a code snippet. We use the task dataset from CodeXGLUE (Lu et al., 2021) which curated a code summarization benchmark from CSN data (Husain et al., 2019). The benchmark consists of six PLs: Ruby, JavaScript, Go, Python, Java, and PHP. It is the same clean version of CSN data that we use for text-to-code retrieval tasks. For training, we set the maximum sequence length of the source and target as 256 and 128, respectively. We use a learning rate of 2e-5,

| Problem | Problem | Problem |
|---|---|---|
| Toulouse has twice as many sheep as Charleston. Charleston has 4 times as many sheep as Seattle. How many sheep do Toulouse, Charleston, and Seattle have together if Seattle has 20 sheep? | Janet's ducks lay 16 eggs per day. She eats three for breakfast every morning and bakes muffins for her friends every day with four. She sells the remainder at the farmers' market daily for $2 per fresh duck egg. How much in dollars does she make every day at the farmers' market? | Eliza's rate per hour for the first 40 hours she works each week is $10. She also receives an overtime pay of 1.2 times her regular hourly rate. If Eliza worked for 45 hours this week, how much are her earnings for this week? |
| **Generated Program** | **Generated Program** | **Generated Program** |
| n0 = 4
n1 = 20
n2 = 2
t0 = n0 * n1
t1 = n2 * t0
answer = t1 + t0 + n1 | n0 = 3
n1 = 4
n2 = 16
n3 = 2
t0 = n0 + n1
t1 = n2 - t0
answer = t1 * n3 | n0 = 10
n1 = 40
n2 = 1.2
n3 = 45
t0 = n0 * n1
t1 = n0 * n2
t2 = n3 - n1
t3 = t2 * t1
answer = t0 + t3 |

Figure 7: Predictions of our model on GSM8K-Python

the batch size as 64 for 10 epochs of finetuning. We set the beam size as 5 in inference.

### E.3 Code Defect Detection

Defect detection is the task of classifying whether a code sample contains vulnerability points or not. We adopt the defect detection benchmark from CodeXGLUE (Lu et al., 2021) which curated data from the Devign dataset (Zhou et al., 2019). The dataset contains in total more than 27,000 annotated functions in C programming language. All samples are collected from popular open-source projects such as QEMU and FFmpeg. We follow (Lu et al., 2021) and adopt 80%/10%/10% of the dataset as the training/validation/test split. For training, we set the learning rate as 2e-5, the batch size as 32, and the max sequence length as 512 to finetune the model for 10 epochs.

### E.4 Code Clone Detection

The task of clone detection aims to detect whether any two code samples have the same functionality or semantics. We conduct experiments using the clone detection benchmark from CodeXGLUE (Lu et al., 2021). The benchmark is curated from the BigClone dataset (Svajlenko et al., 2014) and the resulting curated data consists of 901,724/416,328/416,328 examples for training/validation/test splits respectively. All samples are categorized into 10 different functionalities. For finetuning, we set the learning rate as 2e-5 and finetune the model for 2 epochs. We set the batch size as 10, and the max sequence length as 400.

### E.5 Code Completion

In code completion, given a source sequence containing a partial code sample, a model is required to generate the remaining part of the code sample. We conduct experiments on line-level code completion using two major benchmarks: PY150 (Raychev et al., 2016) and JavaCorpus (Allamanis and Sutton, 2013). PY150 (Raychev et al., 2016) consists of 150,000 Python source files collected from Github. Among these samples, (Lu et al., 2021) selected 10,000 samples from different files from the test set of PY150 and then randomly sampled lines to be predicted for the code completion task. The average numbers of tokens in the source sequence and target sequence are 489.1 and 6.6 respectively. JavaCorpus (Allamanis and Sutton, 2013) contains over 14,000 Java projects collected from GitHub. Similarly to PY150, Lu et al. (2021) selected 3,000 samples from different files from the test set of the dataset and randomly sampled lines to be predicted for the code completion task. The average numbers of tokens in the source sequence and target sequence are 350.6 and 10.5 respectively. For both tasks, we set the learning rate as 2e-5 and batch size as 32, and set the maximum sequence length of 1024 for the decoder. We finetune the model for 30 epochs. During inference, we employ beam search with a beam size of 5.

### E.6 Math Programming

Math Programming is the task of solving maths-based problems with programming. Compared to conventional code generation tasks, this task fo-

| Problem | Problem | Problem |
|---|---|---|
| A shopkeeper sold an article offering a discount of 5% and earned a profit of 31.1%. What would have been the percentage of profit earned if no discount had been offered?
n0 = 5.0 n1 = 31.1 | What will be the difference between simple and compound interest at 14% per annum on a sum of rs. 1000 after 4 years?
n0 = 14.0 n1 = 1000.0 n2 = 4.0 | A full stationary oil tank that is a right circular cylinder has a radius of 100 feet and a height of 25 feet. Oil is pumped from the stationary tank to an oil truck that has a tank that is a right circular cylinder until the truck 's tank is completely filled. If the truck 's tank has a radius of 6 feet and a height of 10 feet, how far (in feet) did the oil level drop in the stationary tank?
n0 = 100.0 n1 = 25.0 n2 = 6.0 n3 = 10.0 |
| **Generated Program** | **Generated Program** | **Generated Program** |
| n0 = 5.0
n1 = 31.1
t0 = n1 + 100.0
t1 = 100.0 - n0
t2 = t0 * 100.0
t3 = t2 / t1
answer = t3 - 100.0 | n0 = 14.0
n1 = 1000.0
n2 = 4.0
t0 = n0 / 100.0
t1 = t0 + 1.0
t2 = n1 * t0
t3 = n2 * t2
t4 = t1**min(n2, 5)
t5 = n1 * t4
t6 = t5 - n1
answer = t6 - t3 | import math
n0 = 100.0
n1 = 25.0
n2 = 6.0
n3 = 10.0
t0 = math.pi * n0**2
t1 = math.pi * n2**2 * n3
answer = t1 / t0 |

Figure 8: Predictions of our model on MathQA-Python

**Question**: Natalia sold clips to 48 of her friends in April, and then she sold half as many clips in May. How many clips did Natalia sell altogether in April and May?
**Answer**:
Natalia sold 48/2 = <<48/2=24>>24 clips in May.
Natalia sold 48+24 = <<48+24=72>>72 clips altogether in April and May.

**Python Solution:**
n0 = 48
n1 = 2
t0 = n0 / n1
answer = n0 + t0

Figure 9: One example of how to convert natural language solution into a Python program on GSM8K dataset.

cuses more on computational reasoning skills. The problem descriptions in this type of task are also more complex than conventional code generation tasks. We employ two major benchmarks for this task: MathQA-Python (Austin et al., 2021) and GradeSchool-Math (Cobbe et al., 2021).

MathQA-Python (Austin et al., 2021) is developed from the MathQA dataset (Amini et al., 2019) where given a mathematical problem description in natural language, a system is required to solve this problem via generating a program that returns the final answer. (Austin et al., 2021) translated these programs into Python programs and filtered for cleaner problems. In total, MathQA-Python contains ∼24,000 problems, including 19,209/2,822/1,883 samples for training/validation/test splits.

GradeSchool-Math (Cobbe et al., 2021) (also known as GSM8K) has similar nature as MathQA. The benchmark focuses on problems with moderate difficulty that an average grade school student should be able to solve. In total, GSM data contains 8,500 problems, divided into 7,500 training and 1,000 testing problems. We translated the

solution described in natural language to Python programs by following the construction process of MathQA-Python by Austin et al. (2021). Finally, we successfully converted 5,861 out of 7,500 training samples. One case can be found in Fig. 9.

For training, we set the maximum sequence length of the source and target as 256 and 256 for MathQA-Python, and 246, 138 for GSM8k-Python. We use a learning rate of 2e-5 and a batch size of 32 for 30 epochs of finetuning. During inference, we employ the beam size as 5 to get pass@1 results. For pass@80 and pass@100, we found they are quite sensitive to the diversity of the generation. We employ nucleus sampling with a temperature of $1.2$ and top-$p$=0.95.

### E.7 Retrieval-augmented Code Generation

Developers often search for relevant code snippets from sources on the web such as GitHub or Stack-Overflow as references to aid their software development process. Motivated by this behaviour, we explore a retrieval-augmented code generation setting, where given a natural language description, a retriever first retrieves similar candidates

in a search codebase and then augments the input for the generator to produce the target code. Such retrieval-augmented generation (or retrieve-then-generate) paradigm has been widely used in open-domain question answering (Karpukhin et al., 2020) in NLP and recently extended to some code-related tasks such as code generation and summarization (Parvez et al., 2021), and program repair tasks (Wang et al., 2023). As our CodeT5+ is capable of both retrieval and generation, it can be seamlessly adapted as a unified retrieval-augmented generator. This can bring unique benefits such as less computational cost compared to prior work that employs a different retriever and generator. We evaluate CodeT5+ on two Java and Python code generation datasets from the CodeXGLUE (Lu et al., 2021) benchmark following Parvez et al. (2021).

Specifically, we leverage the encoder to encode the code snippet in the retrieval base and build a search index with the `faiss` library (Johnson et al., 2019). The search index is a set of representations (of 256 dimensions) for all the code snippets in the retrieval codebase. Let $(x_i, y_i)$ denote one training instance where $x_i$ is the input text description and $y_i$ is the corresponding target code snippet. we employ the same encoder to obtain the embedding of $x_i$ and retrieve top-$k$ similar code samples from the search base using the L-2 similarity metric, with $k$ being a hyperparameter. We ensure that the training example's target string ($y_i$) is not present in any of these $k$ retrieved samples.

After retrieving these top-$k$ relevant code samples, we combine them with a special token `[SEP]` and concatenate it to the end of the source input $x_i$. Unlike (Parvez et al., 2021), we do not augment docstrings or text descriptions and only augment the code snippet for simplicity. We then finetune CodeT5+ on this augmented dataset. During inference, we retrieve similar code samples from the search base and augment these to input $x_i$. For training, we set the maximum sequence length of the source and target as 600 and 320. We use a learning rate of 2e-5, the batch size as 32 to finetune the model for 10 epochs. We set the beam size as 5 during inference with beam search.