# OpenReview forum: "CodeT5+: Open Code Large Language Models for Code Understanding and Generation"
_EMNLP/2023/Conference — EMNLP 2023 Main_

### Official Review · Reviewer_kUr4 · 2023-08-03

**Soundness:** 4

**Excitement:**

4: Strong: This paper deepens the understanding of some phenomenon or lowers the barriers to an existing research direction.

**Paper Topic And Main Contributions:**

The paper argues that existing code Language Models (LLMs) have two main limitations concerning architecture and pretraining tasks. Firstly, they adopt the same fixed architecture for different downstream tasks, resulting in inflexibility in applications and suboptimal performance on specific tasks. Secondly, they often utilize a limited set of pretraining objectives that may not be relevant to certain downstream tasks, leading to substantial performance degradation.
To address these limitations, the authors propose "CodeT5+," a family of encoder-decoder LLMs for code, where component modules can be flexibly combined to suit a wide range of downstream code tasks.Furthermore, they propose initializing CodeT5+ with frozen off-the-shelf LLMs without training from scratch to efficiently scale up their models. They also explore instruction-tuning to align the model with natural language instructions.
They observe state-of-the-art (SoTA) performance on various code-related tasks.

**Questions For The Authors:**

1. This model includes a shallow encoder. Have there been any discussions about which tasks necessarily require the addition of an encoder? For instance, when compared to GPT models of the same size, does the addition of an encoder result in significant performance improvements?

2. How does the masking mechanism of CodeT5+ compare to other dynamic masking techniques such as the UniLM-designed UnixCoder mentioned in the paper, or model like GLM? Beyond experimental outcomes, what is your perspective on these architectures and their training methodologies?

3. I suggest "Unified Retrieval-Augmented Generation" reflected in the abstract. I think it is an interesting aspect of the research.

**Reasons To Accept:**

1. The paper provides a clear summary and analysis of the limitations of existing work. The authors first analyze the limitations of code LLMs adopting a single architecture and then highlight issues with their pretraining tasks, indicating that prior code LLMs may not be perfectly transferable to some downstream tasks.

2. The authors provide a detailed description of CodeT5+ in terms of architecture, pretraining tasks, and computational resources. CodeT5+ adopts Unimodal Pretraining on Code Data and Bimodal Pretraining on Text-code Data, designing different types of pretraining tasks to enhance the model's capabilities. The computationally-efficient pretraining strategy to initialize CodeT5+ with off-the-shelf pre-trained LLMs is crucial and valuable for training large models.

3. The authors select representative datasets for code understanding and code generation tasks, along with different architecture large models, for comprehensive comparisons, demonstrating the effectiveness of CodeT5+. And It is meaningful for the authors to use MathQA and GSM8K as datasets to review code model reasoning capabilities.

**Reasons To Reject:**

1. Are there any experiments to demonstrate the impact of the "Compute-efficient Pretraining with Frozen Off-the-shelf LLMs" method on performance?

2. [1] mentioned that a decoder-only architecture has an advantage over an encoder-decoder architecture in in-context learning, as the prompt can more directly influence the parameters at each layer of the decoder, resulting in a stronger fine-tuning signal. While this may seem like nitpicking, I'm curious if you have compared the differences between CodeT5 in an encoder-decoder configuration and a decoder-only configuration under the same parameter conditions?

[1] "Why can gpt learn in-context? language models secretly perform gradient descent as meta optimizers." arXiv preprint arXiv:2212.10559 (2022).

**Reproducibility:**

4: Could mostly reproduce the results, but there may be some variation because of sample variance or minor variations in their interpretation of the protocol or method.

**Reviewer Confidence:**

4: Quite sure. I tried to check the important points carefully. It's unlikely, though conceivable, that I missed something that should affect my ratings.

**Typos Grammar Style And Presentation Improvements:**

The description of the pretraining tasks is somewhat limited in length, especially regarding "Text-Code Contrastive Learning." The definitions of positive and negative samples are not entirely clear.

---

> ### Author Rebuttal · Authors · 2023-08-28
>
> Thank you for your constructive and positive feedback! We are pleased that you find our paper well motivated by addressing two main limitations of existing work,  our computationally-efficient pretraining strategy crucial and valuable, and our model technically sound and effective demonstrated with comprehensive evaluation.
>
> Below, we provide a detailed response to each of your questions.
>
> # Q1: “Are there any experiments to demonstrate the impact of the "Compute-efficient Pretraining with Frozen Off-the-shelf LLMs" method on performance?”
> The main goal of our computationally-efficient pretraining strategy is to integrate the off-the-shelf GPT LLMs into our CodeT5+’s encoder-decoder architecture to avoid training from scratch. For its impact on performance, we did some comparisons on whether to freeze the deep decoder LLM before and found that for CodeT5+ >= 2B, these two options lead to very comparable results while the unfreezing one consumes much larger GPU memory and also results in much longer training time. We will include more discussions on this in the final version.
>
> # Q2: “I'm curious if you have compared the differences between CodeT5 in an encoder-decoder configuration and a decoder-only configuration under the same parameter conditions?”
> Thanks for the question. We did compare different modes of CodeT5+ before selecting the optimal one for a specific downstream task. Our conclusion is that the decoder-only mode is the optimal mode of CodeT5+ for code completion and the encoder-only mode is the best one for detection tasks, while the encoder-decoder one is the best for Seq2Seq tasks such as text-to-code generation and code summarization, and text-to-code retrieval.
>
> # Q3: “Have there been any discussions about which tasks necessarily require the addition of an encoder? For instance, when compared to GPT models of the same size, does the addition of an encoder result in significant performance improvements?”
> We did not include such exact experiments in the paper. But we do observe that the encoder brings significant improvements for downstream tasks such as text-to-code generation, code summarization, and text-to-code retrieval when selecting the optimal modes for each task. We can also partially observe the benefits of the encoder in the current experiment results such as the text-to-code retrieval in Table 4, where we can find that CodeT5+ 220M attains significant and consistent improvements over CodeGen-multi 350M (decoder-only model) across multiple benchmarks.
>
> # Q4: “How does the masking mechanism of CodeT5+ compare to other dynamic masking techniques such as the UniLM-designed UniXcoder?”
> CodeT5+ does not adopt the dynamic masking mechanism for different downstream tasks like UniXcoder, but instead proposes a flexible encoder-decoder architecture that can operate in different optimal modes for a wide range of understanding and generation tasks.
> Through extensive comparison with UniXcoder, we can verify the effectiveness and advantages of CodeT5+ by observing clear and consistent performance gains from our model (see Table 3,4,7,8). One potential reason for the inferior performance of UniXcoder is that it shares the same Transformer architecture for many different tasks, which might have some inter-task interference and affect its performance on certain tasks. By contrast, our flexible architecture design can reduce such effects and unleash the potential of each component to perform optimally for its corresponding mode.
>
> # Q5: “I suggest "Unified Retrieval-Augmented Generation" reflected in the abstract. I think it is an interesting aspect of the research.”
>
> Thanks for recognizing this interesting aspect of our work! We will add it back to the abstract.
>
> # Q6: “The definitions of positive and negative samples are not entirely clear.”
>
> Thanks for pointing out this. We will make it clearer in the final version. Basically the positive samples in the Text-Code Contrastive Learning refer to text-code pairs where the text is the associated docstrings of the code, which usually specifies the functionality of code. Negative samples refer to text-code pairs where the text and code are not matched in terms of semantics.

---

### Official Review · Reviewer_Ke7W · 2023-08-05

**Paper Topic And Main Contributions:** N/A
**Soundness:** 4

**Excitement:**

4: Strong: This paper deepens the understanding of some phenomenon or lowers the barriers to an existing research direction.

**Reasons To Accept:**

N/A

**Reasons To Reject:**

N/A

**Reproducibility:**

4: Could mostly reproduce the results, but there may be some variation because of sample variance or minor variations in their interpretation of the protocol or method.

**Reviewer Confidence:**

3: Pretty sure, but there's a chance I missed something. Although I have a good feel for this area in general, I did not carefully check the paper's details, e.g., the math, experimental design, or novelty.

---

> ### Author Rebuttal · Authors · 2023-08-28
>
> Thank you for your positive rating. We would love to hear more of your feedback on our work.

---

### Official Review · Reviewer_MqQi · 2023-08-05

**Soundness:** 4

**Excitement:**

4: Strong: This paper deepens the understanding of some phenomenon or lowers the barriers to an existing research direction.

**Paper Topic And Main Contributions:**

To solve two main limitations using LLM, the paper proposes CodeT5+ for code tasks. The CodeT5+ is an encoder-decoder model that can flexibly operate in encoder-only, decoder-only, and encoder-decoder modes to suit different downstream tasks. Evaluations show the effectiveness of the model.

**Reasons To Accept:**

The proposed model gives a new model architecture and training method for code LLM and the model can outperform other LLMs using few parameters. Besides, the training strategy for different tasks is also interesting.

**Reasons To Reject:**

The author should give a clear method to download pre-training data.

**Reproducibility:**

2: Would be hard pressed to reproduce the results. The contribution depends on data that are simply not available outside the author's institution or consortium; not enough details are provided.

**Reviewer Confidence:**

4: Quite sure. I tried to check the important points carefully. It's unlikely, though conceivable, that I missed something that should affect my ratings.

---

> ### Author Rebuttal · Authors · 2023-08-28
>
> Thank you for your positive feedback! We are glad that you recognize the novelty of our proposed model architecture and training strategy, and our effectiveness in outperforming other LLMs using few parameters. Below, we provide response to your concerns of reproducibility:
>
> # Q1: “a clear method to download pre-training data.”
> All of our pre-training data are from existing open-source code corpora including [CodeSearchNet](https://huggingface.co/datasets/code_search_net) and the recently released [GitHub Code](https://huggingface.co/datasets/codeparrot/github-code) dataset. In total, we select nine PLs (Python, Java, Ruby, JavaScript, Go, PHP, C, C++, C#) and filter the dataset by preserving only permissively licensed code. Please refer to the Appendix D.1 for more information.
>
> To further ensure the reproducibility, we will release more instructions for how to preprocess these pretraining data upon the release of this paper. Additionally, code and pretrained checkpoints will be released for replicating the main results of this work.

---

### Meta-Review · Area_Chair_fgaT · 2023-09-17

**Recommendation:** 4

**Metareview:**

This paper addresses the limitations of existing code Language Models (LLMs) by proposing CodeT5+, a flexible encoder-decoder model capable of adapting to a wide range of downstream code tasks. The authors investigate the model's architecture, pretraining tasks, and computational resources, and demonstrate its effectiveness through comprehensive comparisons with other large models. The paper is well-received for its detailed analysis of existing LLM limitations, and the innovative training strategies implemented in CodeT5+.

Strengths:
* CodeT5+ introduces a new model architecture and training method for code LLMs, improving performance with fewer parameters.
* The paper provides a clear summary of the limitations of existing work and offers practical solutions.
* The authors present a detailed description of CodeT5+ and its components, including a computationally-efficient pretraining strategy.
* The model demonstrates state-of-the-art performance on various code-related tasks.

Weaknesses:
* The paper lacks clarity on some details of the paper.
* Some reviewers raised questions about the impact of the "Compute-efficient Pretraining with Frozen Off-the-shelf LLMs" method on * performance and the differences between encoder-decoder and decoder-only configurations.
* The description of pretraining tasks, specifically "Text-Code Contrastive Learning," could be improved for better understanding.

Overall, the paper is well-executed and provides valuable insights into the limitations of existing code LLMs and the potential benefits of the proposed CodeT5+ model. The study is sound and generates excitement in the field by deepening the understanding of code LLMs and lowering barriers to existing research directions. However, some concerns need to be addressed, including the lack of clarity on pre-training data access and the need for more details on certain pretraining tasks.

---

### Decision · Program_Chairs · 2023-10-07

**Decision:**

Accept-Main

**Comment:**

This paper addresses the limitations of existing code Language Models (LLMs) by proposing CodeT5+, a flexible encoder-decoder model capable of adapting to a wide range of downstream code tasks. The authors investigate the model's architecture, pretraining tasks, and computational resources, and demonstrate its effectiveness through comprehensive comparisons with other large models. The paper is well-received for its detailed analysis of existing LLM limitations, and the innovative training strategies implemented in CodeT5+.

Strengths:
* CodeT5+ introduces a new model architecture and training method for code LLMs, improving performance with fewer parameters.
* The paper provides a clear summary of the limitations of existing work and offers practical solutions.
* The authors present a detailed description of CodeT5+ and its components, including a computationally-efficient pretraining strategy.
* The model demonstrates state-of-the-art performance on various code-related tasks.

Weaknesses:
* The paper lacks clarity on some details of the paper.
* Some reviewers raised questions about the impact of the "Compute-efficient Pretraining with Frozen Off-the-shelf LLMs" method on * performance and the differences between encoder-decoder and decoder-only configurations.
* The description of pretraining tasks, specifically "Text-Code Contrastive Learning," could be improved for better understanding.

Overall, the paper is well-executed and provides valuable insights into the limitations of existing code LLMs and the potential benefits of the proposed CodeT5+ model. The study is sound and generates excitement in the field by deepening the understanding of code LLMs and lowering barriers to existing research directions. However, some concerns need to be addressed, including the lack of clarity on pre-training data access and the need for more details on certain pretraining tasks.